# Barriers and Opportunities for Patient-Reported Outcome Implementation: A National Pediatrician Survey in the United States

**DOI:** 10.3390/children9020185

**Published:** 2022-02-02

**Authors:** Ming Chen, Conor M. Jones, Hailey E. Bauer, Onyekachukwu Osakwe, Pavinarmatha Ketheeswaran, Justin N. Baker, I-Chan Huang

**Affiliations:** 1College of Medicine, University of Tennessee Health Science Center, Memphis, TN 38163, USA; mchen46@uthsc.edu; 2Weill Medical College, Cornell University, New York, NY 10021, USA; cmj2002@med.cornell.edu; 3Department of Epidemiology and Cancer Control, St. Jude Children’s Research Hospital, Memphis, TN 38105, USA; Hailey.Bauer@stjude.org; 4School of Medicine, University of Mississippi, Jackson, MS 39216, USA; oosakwe@umc.edu; 5Alpert Brown Medical School, Brown University, Providence, RI 02903, USA; pavinarmatha_ketheeswaran@brown.edu; 6Department of Oncology, St. Jude Children’s Research Hospital, Memphis, TN 38105, USA; Justin.Baker@stjude.org

**Keywords:** barrier, implementation, patient-reported outcomes, pediatrics, physician survey

## Abstract

Purpose: To characterize pediatricians’ perceived barriers and areas of confidence in assessing patient-reported outcomes (PROs) in the U.S., and to test associations of these factors with implementing PRO assessment. Methods: Using a random sample from the members of American Medical Association, we recruited general pediatricians and pediatric subspecialists to complete a survey (July 2011 to December 2013). Perceived barriers and confidence in PRO assessment were compared by age, pediatric specialty (general pediatrics, seven subspecialties), practice settings (academic, private), and region of practice. Multivariable logistic regressions tested associations of demographic factors, barriers, and confidence factors with the implementation of PRO assessment. Findings: The survey was completed by 458 participants (response rate 48.5%); of these, 40.4%, 15.9%, 15.5%, and 8.1% were general pediatricians, cardiology, hematology/oncology, and pulmonary specialists, respectively. PRO assessment was implemented by 29.0% of the pediatricians. The top five barriers for PRO assessment included limited time/manpower (79.0%), limited training (77.4%), lengthy PRO instruments (76%), lack of meaningful cut-offs on PRO scores (75.5%), and unavailable PRO instruments (75.0%). Limited knowledge of PROs (OR 4.10; 95% CI 2.21, 7.60) and unavailability of PRO instruments (OR 1.87; 95% CI 1.01, 3.49) increased the odds of not implementing the assessment, whereas confidence in PRO assessments compatible with norms (OR 0.41; 95% CI 0.23, 0.72) and perceived benefit over clinical judgment alone (OR 0.53; 95% CI 0.31, 0.93) decreased the odds of not implementing the assessment. Interpretation: significant barriers to PRO assessment in pediatric settings suggest the need for providing training, resources, and practical guidance toward implementation. Patient or Public Contribution: healthcare service users contributed to this study by completing a survey and providing feedback about the barriers and areas of confidence in assessing PROs for pediatric populations.

## 1. Introduction

Patients’ symptoms, quality of life, functional status, and life satisfaction (known as “patient-reported outcomes” or “PROs”) are rarely observable but can be measured through self-reports. Various PRO instruments have been developed [1,2], serving as useful tools to track patients’ health status and facilitate clinical decision-making [3]. In addition, PROs are considered as an important endpoint in clinical trials, providing evidence of responsiveness and efficacy for therapeutic regimens [4]. While PRO assessment plays an important role in measuring children’s health status, the use of PRO assessment in pediatric settings is still limited compared to the use of vital signs and physiologically-based laboratory tests. Despite the robust evidence of psychometric and clinical validity in assessing pediatric PROs [5,6], real-world barriers to implementing PROs among pediatric patients are unclear [7].

The literature has documented the advantages of assessing PROs, such as screening patients at risk of health problems and facilitating patient–physician communication [6,8]. In addition, PRO assessment can improve patients’ awareness of symptomatic problems, empower them to discuss concerns with clinicians, and facilitate tailored interventions with proper palliative care support [9,10]. Indeed, PRO assessment has been associated with the improvement of patients’ emotional, social, and psychological functioning status [8,11,12,13]. Nevertheless, physicians have reported barriers to PRO assessment, typically limited time, inadequate PRO collection infrastructure, lack of knowledge of PROs and score interpretations, and skepticism over the validity of PRO assessment [6,14,15,16].

PRO utilization and constraint issues in pediatrics is more complex than in adults given the distinct developmental issues. The assessment of pediatric PROs is hindered by the patient’s younger age and therefore cognitive capability, which may result in the need for caregiver involvement [11,17]. Although few studies have examined the associations between physician characteristics and PRO assessment, these studies are largely restricted to oncologists, radiation oncologists, and urologists, typically in adult settings [18,19]. Only three studies have investigated PRO implementation issues from the viewpoint of pediatricians, the reported barriers including uncertain value of PRO assessment [20], lack of confidence or self-efficacy regarding PRO uses [21], and limited knowledge regarding PRO analysis [22].

Using a sample from a large representative physician database, we conducted a cross-sectional postal mail survey among pediatricians to investigate their preferences, perceived barriers, and confidence in assessing PROs in routine practice. Specifically, we examined pediatricians’ preference for using specific PRO domains in clinical practice, identified the barriers and confidence factors in conducting PRO assessment, and evaluated how demographic, practice, and census region characteristics were associated with the barrier and confidence factors. Finally, we tested associations of barrier and confidence factors with the implementation of PRO assessment in pediatric practice.

## 2. Materials and Methods

### 2.1. Study Design and Participant Identification

A national postal mail survey of general pediatricians and pediatric subspecialists in the U.S. was conducted between July 2011 and December 2013. Using a dataset provided by the American Medical Association (AMA) consisting of a random sample of 1000 general pediatricians and 8488 pediatric subspecialists from four census regions (Midwest, Northeast, South, West), we randomly selected 1000 physicians (400 general pediatricians and 600 subspecialists) to take part in this study. We purposely recruited more subspecialists over the general pediatricians because subspecialists often manage patients who are more severe and require complex interventions compared to general pediatricians. Therefore, it is important to collect sufficient samples of subspecialists to better understand their PRO implementation status. 

Eligible study participants were general pediatricians or pediatric subspecialists who were active in practice in the U.S., either as residents, fellows, or attendees. We focused on seven pediatric subspecialty groups: cardiology, endocrinology, gastroenterology, hematology-oncology, nephrology, pulmonology, and rheumatology. We excluded pediatricians with unverified mailing addresses and those who were retired. Pediatricians were required to complete an informed consent form, as a prerequisite to complete the questionnaire. The Institutional Review Board at the University of Florida approved this study.

### 2.2. Survey Questionnaire

This study created a 60-item survey package for assessing PRO implementation issues. The content included the physician’s demographic characteristics, practice background, interest in important PRO domains for assessment, and perceived barriers and areas of confidence in assessing PROs. Demographic characteristics included attained age, sex, and race/ethnicity. Practice background included years in pediatric practice since residency, practice setting, and proportion of pediatric patients whose PROs were assessed. PRO domains of interest included global, physical, emotional, social and spiritual well-being, pain, sleep, and fatigue.

In the survey package, 9 items measured demographic information, and 16 items measured issues relevant to PRO implementation with aspects of physicians’ knowledge, attitude, confidence, and barriers to implementation. In addition, 1 item captured the current status of implementing PRO assessment. Participants were asked about their interest in particular PRO domains using a binary response category (interested or not interested). A five-point Likert scale (1 = strongly disagree; 5 = strongly agree), was used for items measuring physicians’ confidence in using PRO instruments. Another five-point Likert scale was used to measure physicians’ perceived constraints (1 = extremely; 5 = not at all), as well as the extent of implementation (1 = did not intend to assess PROs; 2 = have thought to assess PROs but unlikely in foreseeable future; 3 = plan to assess PROs; 4 = occasionally assess PROs; 5 = regularly assess PROs). We further categorized the implementation status for PRO assessment as “Yes” if the response scores were 4 or 5, and “No” for otherwise.

### 2.3. Data Collection

We sent a research packet via postal mail to eligible participants, followed by up to five reminders through telephone calls and/or postal mail if they did not respond to the invitation within three months. The research packet comprised the informed consent form, survey questionnaire, and a $20-value gift card as incentive. Participants could choose to return the completed survey along with the signed informed consent forms via postal mail, e-mail (scanned document) or facsimile, or to decline by not returning the survey. 

We replaced selected participants who had invalid mailing addresses (per notifications of postal offices) by other samples randomly selected from the general pediatric or subspecialty group they belonged to. We excluded 55 selected participants from the original sampling frame (N = 1000) due to retired (N = 7), deceased (N = 1) or uncertain (N = 48) status, resulting in 945 confirmed eligible participants in the final statistical analyses.

### 2.4. Statistical Analysis

Chi-square tests were performed to examine the crude associations of important PRO domains for assessment and perceived barriers, respectively, with pediatricians’ age, specialty, practice setting, and census region in which they practiced. Multivariable logistic regressions were used to examine the associations of pediatricians’ perceived barriers and confidence factors, respectively, with the status of PRO assessment through two analytic models. Model 1 tested associations of an individual barrier and confidence factor with the status of implementing PRO assessment, controlling for demographic (age, sex, race), practice (specialty, practice duration and setting), and census region (South, Northeast, Midwest, West) factors. Model 2 used a stepwise backward selection to remove statistically nonsignificant barrier or confidence factors (*p*-value ≥ 0.2) from the analysis, controlling for aforementioned covariates. All analyses were performed using STATA 15. Statistical significance was set at *p*-value < 0.05 (two-sided).

## 3. Results

Table 1 shows the characteristics of the 458 pediatricians who returned the questionnaires (response rate 48.5% among 945 confirmed eligible pediatricians): 72% were non-Hispanic white; 62% were over 40 years old; and 38% practiced in the South region. For clinical background, 40.4% were general pediatricians, 15.9% were cardiologists, 15.5% were hematology/oncology specialists, and 8.1% were pulmonary specialists. In addition, 58.5% had been practicing for at least 10 years, and 55.4% practiced in an academic setting. Approximately 30.0% of pediatricians were currently implementing PRO assessment.

Table 2 shows pediatricians’ interest in specific PRO domains for assessment by age, specialty, and practice setting. Over 50% of pediatricians expressed a common interest across nine PRO domains. Of these, the top five domains commonly endorsed by pediatricians were emotional well-being (74.0%), global well-being (73.1%), physical well-being (67.0%), social well-being (59.4%), and pain (57.6%). Younger pediatricians aged 20 to 40 years had more interest in PRO assessment compared to those aged over 40 years, with statistically significant differences in the domains of global well-being (*p*-value = 0.001), emotional well-being (*p*-value = 0.001), social well-being (*p*-value = 0.002), school activities (*p*-value = 0.004), sleep (*p*-value = 0.005), pain (*p*-value = 0.013), and family functioning (*p*-value = 0.028). Pediatricians’ interest in important PRO domains for assessment also varied by specialty. In contrast to other specialties, pediatric hematology/oncology specialties had more interest in assessing fatigue, pain, sleep, and spiritual well-being (all *p*-values < 0.05), whereas gastroenterology or nephrology specialties had more interest in assessing global well-being, school activities, and family functioning (all *p*-values < 0.05). General pediatricians reported lower interest in all PRO domains compared to other specialties. Pediatricians in academic settings had more interest in assessing all PRO domains (except emotional and social well-beings) compared to those who worked in private settings (all *p*-values < 0.05).

Table 3 shows knowledge of and logistic barriers to PRO assessment by pediatricians’ age, practice setting, and census region. The top five reported constraining factors were limited time and manpower (79.0%), limited training (77.4%), long length of PRO instruments (76.0%), lack of clinically meaningful cut-offs for scoring (75.5%), and lack of appropriate PRO instruments for use (75.0%). By pediatrician characteristics, those aged over 40 years reported more barriers to PRO assessment than aged 20–40 years. The statistically significant barrier factors included skepticism about the validity of PRO instruments (*p*-value = 0.002), inadequate reimbursement incentives for assessing PROs (*p*-value = 0.007), lack of evidence that PRO assessment improves care (*p*-value = 0.008), and unavailability of computerized modes for administering PRO assessment (*p*-value = 0.015). 

Table 3 additionally shows that pediatricians who practiced in the South or Northeast encountered more barriers than those in the Midwest or West. The frequently cited barriers by pediatricians in the South were limited training in PRO instruments (82.0%), lack of clinically meaningful cutoffs (79.7%), and limited skills in scoring PROs (79.5%). There were notable differences in the experience of barriers across census regions, including poor knowledge of PRO concepts, scoring and interpretation, varying capabilities of children, unavailability of computerized modes of administration, lack of evidence that PRO assessment improves care, and skepticism about the validity of PRO instruments (all *p*-values < 0.05). Multivariable analyses suggest that pediatricians who were older (versus younger) in age and resided in the South and Midwest (versus West) encountered more barriers significantly associated with PRO assessment (Appendix A).

Despite these barriers, 44% of pediatricians felt confident that PRO assessment provides more benefits to patients than relying on clinical judgement alone, and 40% of the pediatricians indicated that PRO assessment is compatible with their norms (Figure 1). However, only 26% of the pediatricians were confident in their ability to administer PRO instruments, and 20% expressed concerns about the availability of instruments that could accurately evaluate PROs. Multivariable analyses suggest that pediatricians who practiced in the private (versus academic) setting perceived less benefit and norm compatibility in assessing PROs (Appendix A).

Table 4 shows associations of perceived barrier and confidence factors with currently not implementing PRO assessment, adjusting for the covariates mentioned in the Methods section (Model 1). Significant barrier factors relevant to knowledge issues included limited training in administering PRO instruments (OR 3.29; 95% CI 1.98, 5.47), and insufficient knowledge of PRO concepts (OR 5.80; 95% CI 3.50, 9.60), scoring (OR 3.32; 95% CI 2.03, 5.41) and interpretation (OR 3.04; 95% CI 1.89, 4.93). Significant barriers relevant to logistic or resource issues included unavailability of appropriate PRO instruments (OR 2.90; 95% CI 1.77, 4.74), skepticism about the validity of PRO instruments (OR 2.78; 95% CI 1.72, 4.50), lack of effectiveness that PRO assessment improves care (OR 2.48; 95% CI: 1.55, 3.97), lack of clinically meaningful cutoffs (OR 2.30; 95% CI 1.40, 3.77), and lack of recommendations on follow-up and referral services (OR 2.14; 95% CI 1.33, 3.44).

Using a parsimonious method with a backward variable selection, Table 4 shows significant barrier and confidence factors for not implementing PRO assessment, adjusting for covariates mentioned in the Methods section (Model 2). The significant barrier factors included limited knowledge of PRO concepts (OR 4.10; 95% CI 2.21, 7.60) and unavailability of appropriate PRO instruments (OR 1.87; 95% CI 1.01, 3.49), whereas the identified confidence factors included the assessment’s compatibility with norms (OR 0.41; 95% CI 0.23, 0.72) and perceived benefit over clinical judgment alone (OR 0.53; 95% CI 0.31, 0.93). 

## 4. Discussion

This is one of the largest studies to examine pediatricians’ perceived barriers and areas of confidence associated with the implementation of PRO assessment. Although it is always a challenge to achieve high response rates from physician surveys, the approximately 50% response rate in our study is comparable to previous physician surveys [23,24,25,26,27,28]. Through a comprehensive survey of general pediatricians and pediatric subspecialists across the U.S., we were able to associate physician-related characteristics with the preferences for various PRO domains. In addition, we identified important knowledge, logistic, and resource barriers that affected the implementation of PRO assessment in routine pediatric practice. This study provides robust evidence for the addressing of PRO implementation issues in pediatrics. 

We found that pediatricians who specialized in hematology-oncology, gastroenterology, or nephrology, or who practiced in academic settings, viewed the vast majority of PRO domains as more important versus those who were specialized in general pediatrics, cardiology, pulmonology, and endocrinology, or practiced in private settings. Compared to other specialties, hematologists and oncologists identified pain, fatigue, sleep, and spiritual well-being as important domains because these PRO issues are prevalent in pediatric cancer patients as a result of toxic anticancer therapies and a high stress disease processes [29,30,31,32,33,34]. In addition, hematologists and oncologists were interested in assessing cognitive functioning, which is another side effect resulting from central nervous system-directed therapies, especially prevalent in brain cancer patients [35,36,37,38].

Younger versus older pediatricians reported fewer barriers in PRO assessment. This finding reflects a shift in professional training received in medical education via which the topics of patient centeredness and patient-physician communication have been increasingly emphasized [39]. Younger physicians (including those in training) having fewer barriers against PRO assessment, suggesting that older physicians may have difficulty in instituting new assessment procedures. Expanding educational and organizational efforts (e.g., medical school curriculum, continuous medical education, the development of PRO champions within hospitals) may help to eliminate knowledge constraints and increase health professionals’ confidence or perceived benefits of PRO assessment [39,40].

We found significant variation in geographic locations regarding different barrier factors of PRO implementation. Pediatricians who practiced in the South and Northeast regions reported more barriers compared to those in the West and Midwest. It is likely that varied clinical practice styles or culture, available resources for PRO assessment, and patient characteristics and preferences contribute to the regional differences. The significant barriers reported by pediatricians are in line with previous studies regarding the geographic variation of the assessment of PROs in clinical practice for adults (e.g., constraints in physician education and confidence, clinical organization factors, and specificity of PRO assessments) [40,41,42,43].

Independent of age or geographic location, pediatricians who identified constraints such as limited knowledge of PROs and unavailability of appropriate PRO instruments were significantly less likely to assess PROs. Despite the rapid growth of PRO instruments for use, real-world evidence on the implementation status of PRO assessment is still limited, which requires practical implementation guidance (e.g., integrative PRO data collection platforms and actional referral/treatment plans) [44,45]. We found that, although 45% of the pediatricians acknowledged the usefulness of PRO assessment in providing more benefits than clinical judgment alone, only 26% felt confident in their ability to administer PRO assessment. To remedy this issue, professional PRO or medical societies should recommend actionable, interpretable, and clinically meaningful implementation frameworks for pediatrics [46,47,48]. Special attention should be paid to the aspects of PRO instrument selection, score calculation and interpretation per the needs of pediatric subspecialties and patient populations. For example, the International Society for Quality of Life Research has developed a general guideline, which can be tailored by various pediatric societies for use [44]. Pediatric professional societies should provide recommendations regarding implementation strategies. A recent study found that providing financial incentives significantly increased the rate of PRO assessment [49]. Therefore, health insurance agencies may facilitate appropriate payment mechanisms to optimize PRO data collection, which is now considered an important component of value-based performance programs [50,51].

It is also critical to consider an integrative approach to systematically address barriers in assessing PROs by improving clinical workflow and patient accessibility. Currently, technology-based strategies, e.g., electronic PROs (ePROs), to address PRO implementation issues would be particularly useful as more patients and clinicians utilize mobile and electronic devices. The use of ePROs and computer-adaptive tests (CATs) to monitor PROs can increase the feasibility of PRO assessment remotely, reduce administrative burden, and increase the patient engagement in clinical care [52,53,54,55,56]. Moreover, ePRO platforms allow for tailored care delivery to patients and such information can be linked with electronic health records (EHRs) to facilitate “big-data” initiatives [55,57]. In pediatric settings, PROMIS^®^ and KIDSCREEN have developed free, user-friendly ePROs and CATs. The approach of seamlessly integrating ePROs and CATs with EHRs contributes to real-time clinical decisions.

One of the overlooked implementation issues that can impact PRO integration into clinical workflow is sustainability, especially for longitudinal PRO assessment. This issue is critical for children who are disabled or experience chronic conditions. By establishing meaningful cut-points for PRO domains, developing methods for interpreting score changes over time, and integrating PRO data onto EHRs, we can employ longitudinal data to predict deteriorating health conditions [57]. Developing standardized ePRO platforms will maximize PRO information to be utilized in a variety of care settings and timepoints [47,48,52,53,56]. 

This study has some weaknesses. First, representativeness of our findings may be limited. Our surveys were collected between 2011 and 2013, and the results may not reflect the contemporary practice of pediatric PRO assessments because modern technology (e.g., mobile technology for PRO data collection, integrating PRO reporting into electronic health record systems) may have been used to overcome some PRO implementation barriers in pediatric oncology. Second, our results may not be generalized to all pediatricians in the U.S. because the participating pediatricians were randomly selected from AMA’s members. Future studies are warranted to confirm our findings by using other professional databases (e.g., American Academy of Pediatrics).

In summary, wide variations in the implementation of PRO assessment were found among a national sample of pediatricians. Pediatricians’ preferences in PRO domains significantly differed by age, specialty, and practice setting. Barriers faced by pediatricians varied mostly by attained age and census region, rather than the health systems in which they were affiliated. To overcome these barriers, targeted interventions in disseminating education and training in PRO topics, developing useful PRO cut-points, promoting ePROs, professional society advocacy, advancing reimbursement strategies, and integrating ePROs with EHRs to deliver tailored longitudinal care to children are warranted.

## Figures and Tables

**Figure 1 children-09-00185-f001:**
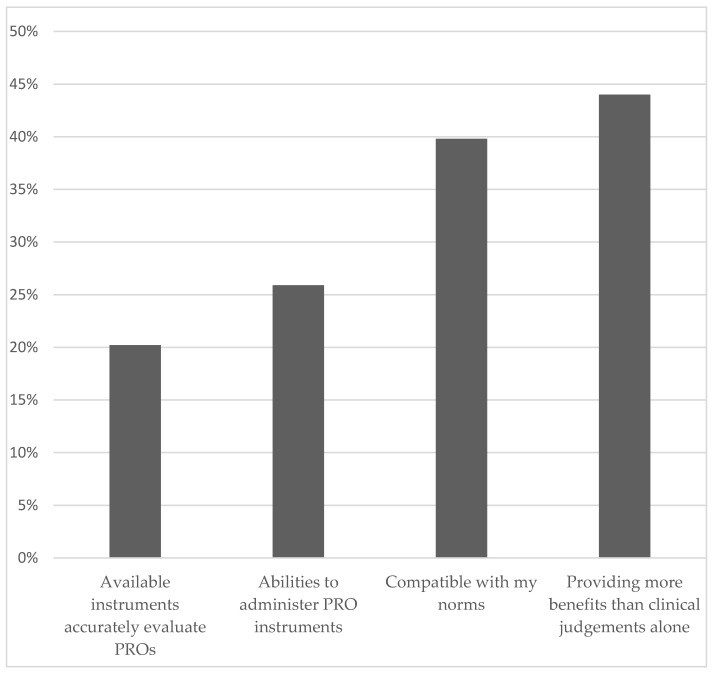
Areas of Confidence in PRO Assessment.

**Table 1 children-09-00185-t001:** Characteristics of Participating Pediatricians (N = 458).

Characteristics	N (%)
**Age in years**	
20–40	174 (38.0)
≥41	284 (62.0)
**Sex**	
Female	220 (48.6)
Male	233 (51.4)
**Race/ethnicity**	
White, non-Hispanic	327 (71.9)
Asian or Pacific Islander	78 (17.1)
Hispanic	25 (5.5)
Black, non-Hispanic	13 (2.9)
Other	12 (2.6)
**Years in pediatric practice since residency**	
0–10	188 (41.5)
≥11	265 (58.5)
**Specialty**	
General Pediatrics	185 (40.4)
Pediatric Cardiology	73 (15.9)
Pediatric Hematology Oncology	71 (15.5)
Pediatric Pulmonology	37 (8.1)
Pediatric Endocrinology	36 (7.9)
Pediatric Gastroenterology	34 (7.4)
Pediatric Nephrology	18 (3.9)
Pediatric Rheumatology	4 (0.9)
**Practice setting**	
Academic	251 (55.4)
Private Practice	202 (44.6)
**U.S. Census Region**	
South	173 (37.8)
Northeast	101 (22.1)
Midwest	99 (21.6)
West	85 (18.6)
**Proportion of patients with PROs being assessed**	
<10	321 (71.2)
≥11	130 (28.8)
**Implementing PRO assessment**	
Currently not being implemented ^†^	327 (71.0)
Currently being implemented ^‡^	127 (29.0)

^†^ Did not intend to assess PROs, have thought to assess PROs but unlikely in foreseeable future, or plan to assess PROs. ^‡^ Regularly or occasionally assess PROs.

**Table 2 children-09-00185-t002:** Important PRO Domains for Assessment Associated with Pediatricians’ Age, Specialty, and Practice Setting ^†^.

	Overall	Age in Years			Specialty ^‡^						Practice Setting		
		20–40	≥41	X^2^ (*p*-Value)	GP	C/P	E/R	HO	G/N	X^2^ (*p*-Value)	Academic	Private	X^2^ (*p*-Value)
PRO Domains	N (%)	N (%)	N (%)		N (%)	N (%)	N (%)	N (%)	N (%)		N (%)	N (%)	
Emotional well-being	339 (74.0)	144 (83.8)	195 (68.7)	11.15 (0.001)	137 (74.1)	79 (71.8)	33 (82.5)	50 (70.4)	40 (76.9)	2.45 (0.648)	195 (77.7)	141 (69.8)	3.64 (0.057)
Global well-being	335 (73.1)	146 (82.2)	192 (67.6)	11.67 (0.001)	120 (64.9)	86 (79.2)	32 (80.0)	53 (74.6)	44 (84.6)	12.40 (0.015)	203 (80.9)	130 (64.4)	15.69 (<.001)
Physical well-being	307 (67.0)	118 (67.8)	189 (66.5)	0.08 (0.780)	111 (60.0)	81 (73.6)	29 (72.5)	53 (74.6)	33 (63.5)	9.02 (0.061)	180 (71.7)	126 (62.4)	4.45 (0.035)
Social well-being	272 (59.4)	119 (68.4)	153 (53.9)	9.43 (0.002)	108 (58.4)	61 (55.5)	27 (67.5)	43 (60.6)	33 (63.5)	2.27 (0.686)	159 (63.3)	112 (55.4)	2.91 (0.088)
Pain	264 (57.6)	113 (64.9)	151 (53.2)	6.13 (0.013)	87 (47.0)	58 (52.7)	26 (65.0)	56 (78.9)	37 (71.2)	27.51 (<0.001)	170 (67.7)	93 (46.0)	21.62 (<0.001)
Family functioning	257 (56.1)	109 (62.6)	148 (52.1)	4.86 (0.028)	91 (49.2)	58 (52.7)	25 (62.5)	46 (64.8)	37 (71.2)	11.72 (0.020)	153 (61.0)	102 (50.5)	4.98 (0.026)
Cognitive functioning	250 (54.6)	105 (60.3)	145 (51.1)	3.76 (0.053)	90 (48.6)	66 (60.0)	22 (55.0)	46 (64.8)	26 (50.0)	7.36 (0.118)	154 (61.4)	94 (46.5)	9.92 (0.002)
School activities	242 (52.8)	107 (61.5)	135 (47.5)	8.44 (0.004)	82 (44.3)	60 (54.5)	23 (57.5)	44 (62.0)	33 (63.5)	10.59 (0.032)	149 (59.4)	91 (45.0)	9.20 (0.002)
Sleep	241 (52.6)	106 (60.9)	135 (47.5)	7.75 (0.005)	81 (43.8)	58 (52.7)	20 (50.0)	48 (67.6)	34 (65.4)	15.70 (0.003)	154 (61.4)	87 (43.1)	15.03 (<0.001)
Fatigue	215 (46.9)	86 (49.4)	129 (45.4)	0.69 (0.405)	63 (34.1)	54 (49.1)	20 (50.0)	52 (73.2)	26 (50.0)	32.60 (<0.001)	141 (56.2)	74 (36.6)	17.14 (<0.001)
Spiritual well-being	121 (26.4)	50 (28.7)	71 (25.0)	0.78 (0.379)	35 (18.1)	31 (28.2)	15 (37.5)	29 (39.4)	12 (23.1)	14.54 (0.006)	79 (31.5)	42 (20.8)	6.52 (0.011)

^†^ Values represents the number of participating pediatricians who agreed that specific PRO domain was important for assessment. **^‡^** Specialty: GP = General Pediatrics; C/P = Pediatric Cardiology or Pulmonology; E/R = Pediatric Endocrinology or Rheumatology; HO = Pediatric Hematology Oncology; G/N = Pediatric Gastroenterology or Nephrology.

**Table 3 children-09-00185-t003:** Bivariate Analyses for Pediatricians’ Age, Practice Setting, and Census Region Associated with Human and Logistic Barriers to PRO Assessment ^†^.

	Overall	Age in Years			Practice Setting			Census Region				
		20–40	≥41	X^2^ (*p*-Value)	Academic	Private Practice	X^2^ (*p*-value)	Northeast	Midwest	South	West	X^2^ (*p*-Value)
Constraints	N (%)	N (%)	N (%)		N (%)	N (%)		N (%)	N (%)	N (%)	N (%)	
Limited time and manpower for assessing PROs	338 (79.0)	130 (77.4)	208 (80.0)	0.42 (0.516)	193 (80.4)	141 (77.0)	0.71 (0.400)	81 (85.3)	81 (83.5)	121 (76.1)	55 (71.4)	6.89 (0.075)
Limited training on how to administer PRO instruments	333 (77.4)	133 (79.2)	200 (76.3)	0.47 (0.493)	185 (76.8)	146 (79.3)	0.41 (0.525)	75 (78.9)	73 (75.3)	132 (82.0)	53 (68.8)	5.56 (0.135)
Long length of PRO instruments	323 (76.0)	129 (76.8)	194 (75.5)	0.09 (0.759)	184 (76.7)	136 (75.6)	0.07 (0.791)	76 (80.9)	73 (76.0)	125 (79.1)	49 (63.6)	8.51 (0.037)
Lack of clinically meaningful cut-offs for PRO scores	321 (75.5)	122 (73.5)	199 (76.8)	0.61 (0.435)	179 (74.9)	139 (76.8)	0.20 (0.653)	75 (79.8)	72 (75.0)	126 (79.7)	48 (62.3)	9.71 (0.021)
Unavailability of appropriate PRO instruments	318 (75.0)	124 (74.3)	194 (75.5)	0.08 (0.774)	182 (76.2)	133 (73.9)	0.28 (0.596)	75 (79.8)	70 (72.9)	121 (77.1)	52 (67.5)	4.02 (0.259)
Limited skills on scoring PRO results	316 (73.7)	119 (71.3)	197 (75.2)	0.81 (0.367)	174 (72.5)	140 (76.1)	0.70 (0.404)	72 (76.6)	66 (68.0)	128 (79.5)	50 (67.9)	7.85 (0.049)
Limited ability to interpret PRO results	307 (71.7)	116 (69.5)	191 (73.2)	0.70 (0.405)	170 (70.8)	135 (73.8)	0.45 (0.505)	70 (74.5)	67 (69.1)	125 (78.1)	45 (58.4)	10.62 (0.014)
Limited knowledge of PRO concepts	309 (71.2)	120 (71.0)	189 (71.3)	0.01 (0.944)	173 (71.5)	133 (71.1)	0.01 (0.934)	72 (75.8)	67 (68.4)	124 (76.5)	46 (58.2)	10.10 (0.018)
Lack of recommendations on follow-up and referral services	299 (70.5)	116 (69.5)	183 (71.2)	0.15 (0.700)	165 (69.0)	131 (72.8)	0.69 (0.405)	72 (76.6)	63 (65.6)	117 (74.5)	47 (61.0)	7.31 (0.063)
Varying capabilities of children	284 (66.8)	108 (64.7)	176 (68.2)	0.58 (0.448)	159 (66.5)	121 (66.9)	0.01 (0.944)	69 (73.4)	54 (56.3)	116 (73.4)	45 (58.4)	12.22 (0.007)
Unavailability of computerized mode for administering PROs	249 (58.8)	86 (51.5)	163 (63.4)	5.94 (0.015)	134 (56.1)	111 (61.7)	1.33 (0.250)	61 (64.9)	41 (42.7)	100 (63.7)	47 (61.0)	13.41 (0.004)
Lack of reimbursement incentives for assessing PROs	241 (56.6)	51 (48.5)	160 (61.8)	7.28 (0.007)	129 (54.0)	110 (60.4)	1.76 (0.185)	58 (61.7)	56 (57.7)	90 (57.0)	37 (48.1)	3.35 (0.341)
Lack of evidence that PRO assessment improves care	220 (51.8)	73 (43.7)	147 (57.0)	7.14 (0.008)	115 (48.1)	103 (56.9)	3.19 (0.074)	55 (58.5)	43 (44.8)	95 (60.1)	27 (35.1)	16.61 (0.001)
Skepticism about the validity of PRO instruments	208 (49.1)	66 (39.5)	142 (55.3)	10.03 (0.002)	110 (46.0)	96 (53.3)	2.19 (0.139)	50 (53.2)	46 (47.9)	89 (56.7)	23 (29.9)	15.69 (0.001)

^†^ See Appendix A for the Results of Multivariable Analyses.

**Table 4 children-09-00185-t004:** Multivariable Analyses for the Barrier and Confidence Factors Associated with Currently Not Implementing PRO Assessment in Clinical Practice ^&^.

Factors	Model 1 ^†^ Regular Multivariable Logistic Regression for Currently not Implementing PRO Assessment	Model 2 ^‡^ Stepwise Backward Multivariable Logistic Regression for Currently not Implementing PRO Assessment
	OR (95% CI)	OR (95% CI)
**Constraint factors**		
Limited time and manpower for assessing PROs	1.98 (1.18, 3.31) *	NS
Limited training on how to administer PRO instruments	3.29 (1.98, 5.47) ***	NS
Long length of PRO instruments	1.66 (1.01, 2.74) *	NS
Lack of clinically meaningful cut-offs for PRO scores	2.30 (1.40, 3.77) **	NS
Unavailability of appropriate PRO instruments	2.90 (1.77, 4.74) ***	1.87 (1.01, 3.49) *
Limited skills on scoring PRO results	3.32 (2.03, 5.41) ***	NS
Limited ability to interpret PRO results	3.04 (1.89, 4.93) ***	NS
Limited knowledge of PRO concepts	5.80 (3.50, 9.60) ***	4.10 (2.21, 7.60) ***
Lack of recommendations on follow-up and referral services	2.14 (1.33, 3.44) **	NS
Varying capabilities of children	1.17 (0.73, 1.87)	NS
Unavailability of computerized mode for administering PROs	1.83 (1.16, 2.90) *	NS
Lack of reimbursement incentives for assessing PROs	1.38 (0.88, 2.16)	NS
Lack of evidence that PRO assessment improves care	2.48 (1.55, 3.97) ***	NS
Skepticism about the validity of PRO instruments	2.78 (1.72, 4.50) ***	NS
**Confidence factors**		
More benefits of PRO assessment than clinical judgments alone	0.32 (0.20, 0.50) ***	0.53 (0.31, 0.93) *
PRO assessment compatible with my norms	0.21 (0.13, 0.33) ***	0.41 (0.23, 0.72) **
Abilities to administer PRO instruments	0.18 (0.11, 0.29) ***	NS
Available instruments accurately evaluate PROs	0.22 (0.13, 0.37) ***	NS

OR = odds ratio; CI = confidence interval; NS = variables not selected into the final mode per stepwise approach (*p*-value ≥ 0.2). * *p*-value < 0.05; ** *p*-value < 0.01; *** *p*-value < 0.001. ^&^ Status of implementation: currently not implementing PRO assessment (do not intend to assess PROs, have thought to assess PROs but unlikely in foreseeable future, or plan to assess PROs) and currently implementing PRO assessment (regularly or occasionally assess PROs). ^†^ Multivariable logistic regression model estimated ORs for PRO assessment currently not being implemented (vs. implemented) associated with individual constraint and confidence variables by adjusting for pediatricians’ demographic, practice and census region covariates. ^‡^ Stepwise backward multivariable logistic regression model estimated ORs for PRO assessment currently not being implemented (vs. implemented) associated with all constraint and confidence variables by adjusting for pediatricians’ demographic, practice and census region covariates.

## Data Availability

Data is available upon request from the corresponding author.

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
