# Peer review of "Barriers and Opportunities for Patient-Reported Outcome Implementation: A National Pediatrician Survey in the United States"

_children, 2022, doi:10.3390/children9020185_

Round 1

Reviewer 1 Report

Authors did a research on the rate of patient-reported outcome use among pediatricians and reported limitations of PRO implementation in the practice. Limited knowledge and unavailability of the instruments were shown as significant barriers to PRO use. Proposed actions are training on PRO, practical guidelines, and increasing resources. Results show interesting preferences towards specific health domains in regard to the demographic or expertise of the participants.
PROs play critical role in assessing the impact of the disease on daily functioning, providing more sensitive data in regard to routinely used biomarkers. PROs also give insight how general, healthy population is functioning, enabling comparison with the patients. Their role in clinical studies is remarkable.
Advantages of this manuscript are:
1) randomization of the participants
2) robust survey questionnaire with intelligent scale
3) reported folowup on non-responders and clear participant enrollment
4) strong statistics
5) clear discussion of the results and thorough comparison with the published literature. I found it important that authors proposed practical guidelines how to systematically address the issues identified with the implementation of PROs in practice.

Please check author list, I believe there is an error for the last listed author ("and" should be omitted).

Author Response

Please see point-by-point response in the attached file. 

Reviewer 2 Report

This is an interesting manuscript. However it is not clear in the methods section the reason to include a sample with general pediatrician / subspecialist pediatrician ratio of 1 to 1.5, when thwy have a source population with a ratio of 1 to 8. In addition, there is not a mention about the low power for comparisons stratifying by subspecialty.

Other point that should be considered is de 10 years delay to publish those results, that may introduce a technology gap, impairing to genarate recommendations to the actual scenary

Author Response

(The authors gave the same response as above.)
